# ACE2 *N*-glycosylation modulates interactions with SARS-CoV-2 spike protein in a site-specific manner

Ayana Isobe[1,8], Yasuha Arai[1,8], Daisuke Kuroda [2,8], Nobuaki Okumura[3], Takao Ono[4], Shota Ushiba [5], Shin-ichi Nakakita [6], Tomo Daidoji[1], Yasuo Suzuki[7], Takaaki Nakaya [1], Kazuhiko Matsumoto[4] & Yohei Watanabe [1✉]

SARS-CoV-2 has evolved continuously and accumulated spike mutations with each variant having a different binding for the cellular ACE2 receptor. It is not known whether the interactions between such mutated spikes and ACE2 glycans are conserved among different variant lineages. Here, we focused on three ACE2 glycosylation sites (53, 90 and 322) that are geometrically close to spike binding sites and investigated the effect of their glycosylation pattern on spike affinity. These glycosylation deletions caused distinct site-specific changes in interactions with the spike and acted cooperatively. Of note, the particular interaction profiles were conserved between the SARS-CoV-2 parental virus and the variants of concern (VOCs) Delta and Omicron. Our study provides insights for a better understanding of the importance of ACE2 glycosylation on ACE2/SARS-CoV-2 spike interaction and guidance for further optimization of soluble ACE2 for therapeutic use.

[1] Department of Infectious Diseases, Kyoto Prefectural University of Medicine, Kyoto 602-8566, Japan. [2] Research Center for Drug and Vaccine Development, National Institute of Infectious Diseases, Tokyo 162-8640, Japan. [3] Institute for Protein Research, Osaka University, Osaka 565-0871, Japan. [4] SANKEN, Osaka University, Osaka 567-0047, Japan. [5] Murata Manufacturing Co., Ltd., Kyoto 617-8555, Japan. [6] Division of Functional Glycomics, Kagawa University, Kagawa 761-0793, Japan. [7] Department of Medical Biochemistry, School of Pharmaceutical Sciences, University of Shizuoka, Shizuoka 422-8526, Japan. [8] These authors contributed equally: Ayana Isobe, Yasuha Arai, Daisuke Kuroda. ✉email: nabe@koto.kpu-m.ac.jp

Two-thirds of animal viruses possess a membrane similar to that of the host cell[1]. Surface proteins on both cellular and viral membranes are glycosylated by the host cellular glycosylation apparatus; these glycans are extremely diverse and highly species-specific[2]. It is known that viruses utilize glycans in various ways when infecting a host cell[3]. Some viruses use glycans on host cells to increase the possibility of viral adsorption onto the host cell, whereas other viruses use glycans as viral entry receptors.

Severe acute respiratory syndrome coronavirus 2 (CoV2), which is responsible for coronavirus disease 19 (COVID-19) is causing an ongoing global health problem[4–6]. Although the Delta variant (B.1.617.2 lineage) had dominated previous variants and had spread widely to >180 countries by September 2021, the newly emerging Omicron variant (B.1.1.529.2/BA.2 lineage) outcompeted the Delta variant and swiftly became the dominant strain worldwide as of January 2022[7]. CoV2 utilizes the human enzyme angiotensin-converting enzyme 2 (hACE2) as a cell receptor[8,9]. The SARS-CoV-2 spike protein (CoV2-S) is trimeric, with each monomer consisting of S1 and S2 subunits[9,10]. S1 includes the receptor-binding domain (RBD) responsible for the virus-host cell interaction, whereas S2 mediates membrane fusion activity[11–13]. CoV2-S is also the principal target of neutralizing antibodies elicited by the infection[14,15]. Since its emergence, CoV2 has evolved continuously, accumulating multiple spike mutations as an adaptation mechanism to escape from immune selection pressure[16]. For example, the spike amino acid D614G mutation was rare before March 2020 but was noted to be increasing in frequency in April 2020. The D614G mutation confers an advantage for infectivity by increasing the stability of virions[17] and inducing structural changes that favor hACE2 binding[18]. In contrast, N501Y, S477N, and N439K on the RBD not only increase hACE2 affinity[19–21] but also are mutations that allow escape from serum antibodies[21,22]. CoV2 variants including Delta and the Omicron lineage emerged with multiple mutations and reportedly exhibit differences in infectivity in vitro and in vivo[23,24] and in hACE2 affinity in silico[25].

Both CoV2-S and the hACE2 receptor are heavily glycosylated[26–28]. Molecular dynamic (MD) simulations of the complexity of their glycosylated forms indicated that glycans on the two molecules may have important implications for hACE2-CoV2 S interactions[28–31]. However, other studies had a different interpretation and postulated that glycosylation does not significantly impact their interaction[32,33]. Another study using CRISPR-Cas9 glycoengineered cells showed that blocking glycan biosynthesis at the oligomannose stage changed CoV2-S binding to hACE2[34]. However, a complete understanding of the role of individual glycans in CoV-2 infection remains to be achieved. In particular, contributions of site-specific glycosylation of hACE2 to interactions with CoV2-S and their relevance for CoV2 infection remain to be defined. Moreover, it is not known whether hACE2 glycan-RBD interactions are conserved among different CoV2 variant lineages or whether they are different for each lineage.

CoV2 vaccines based on novel messenger RNA technology have been licensed for human use[35,36]. Additionally, hACE2 peptide-mimics that block CoV2 infection have been investigated as therapeutic agents[37–39]. Many groups have attempted to optimize soluble hACE2 (shACE2) for increased inhibitory activity, including the addition of multimerization domains[37,38,40], glyco-engineering[41], introduction of mutations found in ACE2 orthologues of different animal species[42], and deep mutagenesis followed by screening[37–39]. However, few studies have systematically assessed the effect of hACE2 glycosylation on shACE2 affinity and inhibitory activity[41], and they were limited to the SARS-CoV-2 parental strain. Thus, the glycosylation effect should be investigated using more SARS-CoV-2 strains including VOCs. In the present study, we investigated the effects of site-specific hACE2 glycosylation on its binding to CoV2-S using recombinant hACE2 glycan variants. Here, we demonstrate that hACE2 glycans located at the interface with the CoV2-S RBD make distinct site-specific contributions to CoV2-S interactions and their net effect determines the efficacy of virus entry into the host cell. We found similar effects during virus adsorption of CoV2 Parental, Delta and Omicron lineages. Our experimental findings will guide the rational design of therapeutic shACE2 with improved blocking efficacy by optimizing their glycosylation pattern to achieve the strongest interaction between the two molecules.

## Results

**Soluble hACE2 variants deficient in *N*-glycosylation at specific sites.** hACE2 has seven canonical sequons for *N*-glycosylation (N-X-S/T, where X is any amino acid except P; Fig. 1a)[26,28]. Among the predicted *N*-glycosylation sites, in this study we focused on N53, N90 and N322, which are geometrically closest to the RBD (Fig. 1b), and investigated the effect of *N*-glycan deletions of the three sites separately and together. To this end, we introduced N to A mutations at the first amino acid (N) of these sequons singly or in combination. In addition, N to S mutants, where S is more similar to N than A, were included to eliminate the possible effects of amino acid substitution per se on the interaction between hACE2 and CoV2-S, as observed in previous amino acid substitutions for hACE2-T324[37].

To initially assess whether N53, N90 and N322 were *N*-glycosylated, shACE2 wild-type (wt) and glycan deletants were purified from the supernatants of cultured human HEK293 cells ectopically expressing them following plasmid transfection. These shACE2 forms encompass a large portion of the ectodomain (19-619 amino acids) of hACE2 and a fused twin-strep-tag at the C-termini (Fig. 1a). Coomassie blue-stained SDS-PAGE exhibited single bands for all shACE2 forms, indicating that a highly purified preparation had been created using the Strep-Tactin XT system (Fig. 2a, b). The wt molecules were extremely heavily glycosylated with their apparent molecular mass being greater than their theoretical mass based on peptide sequences alone (~85 kDa vs 70 kDa). The glycan deletants migrated faster than the wt molecule, depending on the amount of introduced mutations, implying glycan loss at each site. MALDI-TOF mass spectrometry confirmed the glycan deficiency of the deletants, which had smaller average molecular weights (AMW) than the wt molecule, in a deletion-dependent manner (AMW = wt > single > double > triple glycan deletants) (Fig. 2c–e). These data demonstrate that *N*-glycans occupied N53, N90, and N322 and had then been deleted as expected.

**Site-specific *N*-glycan deletions on hACE2 alter its binding affinity to the SARS-CoV-2 Spike RBD.** Using Bio-Layer Interferometry (BLI), we next evaluated the binding of shACE2 wt and the glycan deletants to RBD from the parental Wuhan strain (termed here Parental virus), the Delta (B.1.617.2) lineage (termed here Delta) and the Omicron (B.1.1.529.2/BA.2) lineage (termed here Omicron). Recombinant RBD-Fc was immobilized onto anti-human IgG Fc capture sensors and tested for binding to titrated concentrations of shACE2. Sensorgrams for RBD derived from Parental, Delta and Omicron are shown in Fig. 3a–c respectively. The $K_D$, on-rate and off-rate were calculated by global fitting to a 1:1 steady-state binding model and expressed as the average of two analytical repeats (Fig. 3d–f).

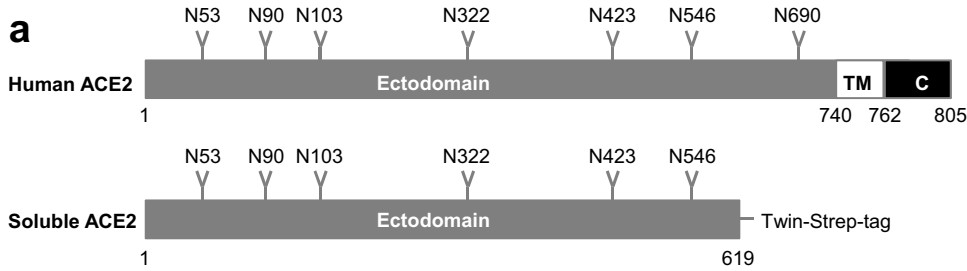

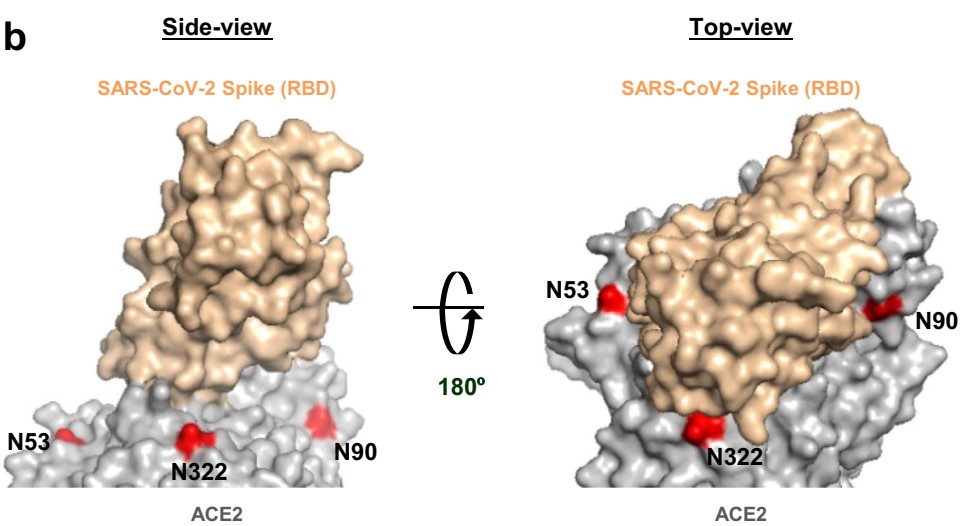

**Fig. 1 hACE2 glycosylation deletions. a** Schematic of the full-length hACE2 protein domain showing potential *N*-glycosylation positions on the soluble hACE2 used in this study. **b** *N*-glycosylation sites (53, 90, and 322) mapped on to the structure of the hACE2/CoV2 S-RBD (PDB: 6VW1) and colored red. (Left) side-view and (right) top-view. The three glycosylation sites positioned close to the hACE2/CoV2 S-RBD interface were deleted singly or in combination.

The shACE2 glycan variants perceptibly changed the binding affinity of Parental RBD up to about 2-fold higher or lower than wt shACE2 (Fig. 3d, left). In particular, the N90A deletant increased the binding affinity ($K_D$: $1.45 \times 10^{-7}$ M vs $3.17 \times 10^{-7}$ M), whereas the N53A deletant reduced it to the lowest value observed in this study ($K_D$: $4.85 \times 10^{-7}$ M). By contrast, the N322A deletant had a less appreciable effect on binding, despite its slightly increased affinity ($K_D$: $2.41 \times 10^{-7}$ M). Combining each glycan deletion resulted in an effect on RBD binding in an additive or offsetting manner: the N90A/N322A double deletant exhibited the highest affinity in this study ($K_D$: $1.15 \times 10^{-7}$ M), whereas the N53A/N90A/N322A triple deletant only moderately increased affinity ($K_D$: $2.54 \times 10^{-7}$ M). These results directly reflected the net effect caused by the reduction mediated by N53A, the increase by N90A and/or the slight increase by N322A. No synergistic effect to further reduce affinity were observed for combined glycan deletants, because only N53A had a reducing effect and the other two deletants had increasing effects.

Although the RBDs from Delta and Omicron had an inherently higher affinity for shACE2-wt than did the Parental RBD ($K_D$: $2.50 \times 10^{-7}$ M for Delta and $1.22 \times 10^{-7}$ M for Omicron vs $3.17 \times 10^{-7}$ M for Parental) (Fig. 3d, middle and right), the three RBDs showed similar binding profiles with the shACE2 glycan variants, such that the variation relative to ACE2-wt was the same for each strain. These variations were accompanied by changes in the on-rate and off-rate caused by glycan deletions (Fig. 3e, f). Furthermore, the N to S variants had effects similar to the N to A variants, indicating that the altered

affinity was due to glycan deficiency and not to amino acid substitutions per se.

### *N*-glycan deletions on hACE change the efficacy of entry of SARS-CoV-2 into host cells in a site-specific manner. To evaluate the biological relevance of different Spike affinities to shACE2 glycan variants for CoV2 infection, HEK293 cells that produce endogenous hACE2 at submarginal levels were made to ectopically express full-length hACE2 wt or the glycan deletants, and were then infected with SARS-CoV-2 Parental, Delta or Omicron. Western blotting using the cell membrane fraction confirmed that hACE2 wt and the glycan deletants were expressed at similar levels on the surface of these HEK293 cells (Fig. 4a, b). Virus entry into these cells by SARS-CoV-2 infection was then evaluated by counting foci, each of which were derived from a single infecting virus particle, in immunofluorescence assays.

The N90A deletant increased the entry of Parental virus during adsorption by 1.71-fold that of the wt, whereas the N53A deletant reduced it by 0.72-fold, the lowest in this study (Fig. 4c). The N322A deletant slightly increased the uptake of the Parental virus by 1.17-fold. Multiple glycan deletions changed virus uptake in an additive or offset manner, as seen in the BLI analyses. Thus, entry was highest for the N90A/N322A double deletant at 1.95-fold that of the wt, but with an intermediate increase for the N53A/N90A/N322A triple deletant of 1.62-fold.

Delta and Omicron showed virus entry profiles relative to hACE2 glycan variants that were indistinguishable from the

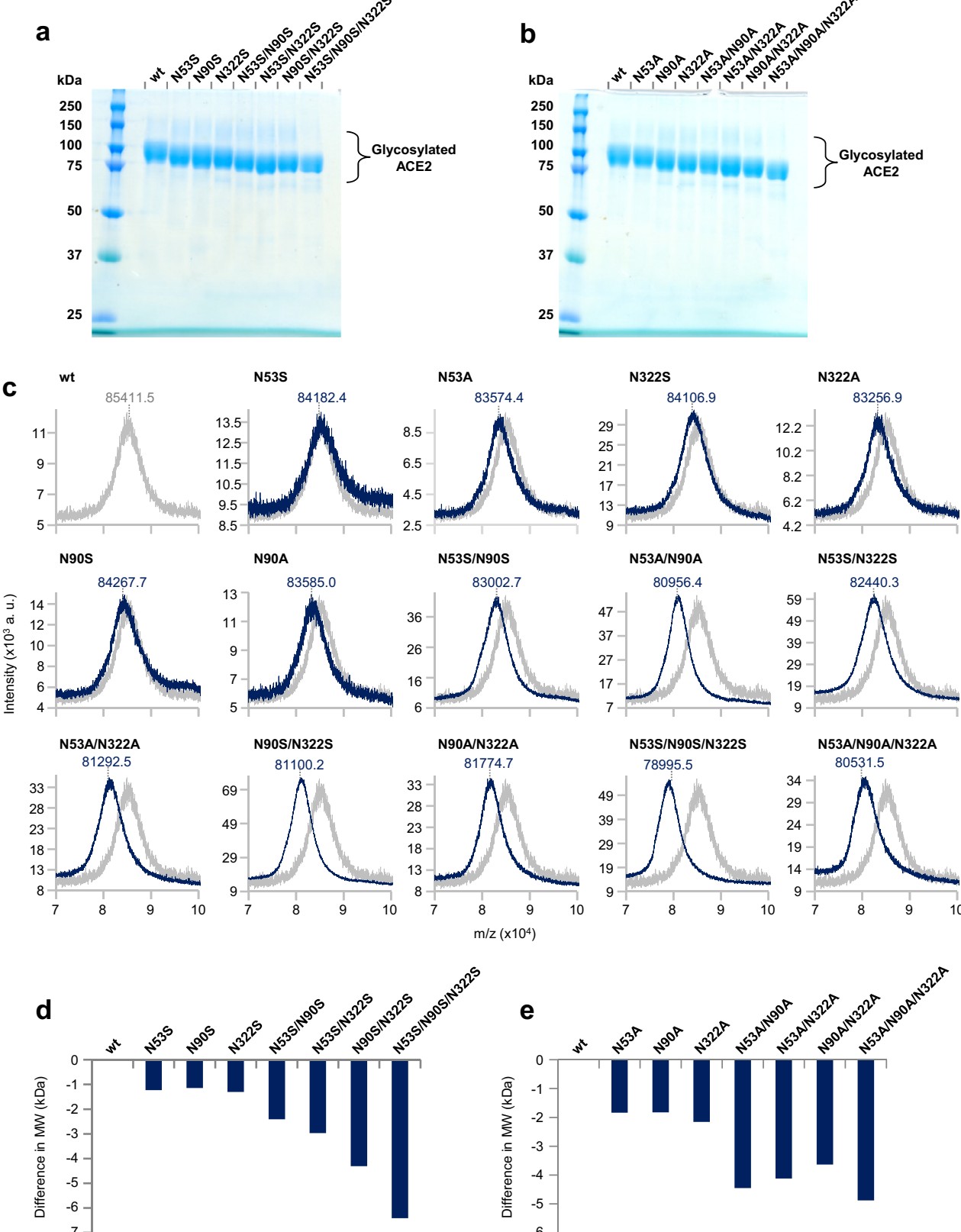

**Fig. 2 Preparation of soluble hACE2 glycan deletants. a, b** SDS-PAGE with Coomassie blue staining of purified recombinant shACE2 wt and the glycan deletants for N-to-S substitution (**a**) and N-to-A substitution (**b**). **c** MALDI-TOF mass spectra of shACE2 wt (light gray) and the glycan deletants (dark blue). Mass spectra were obtained in the range from 15,000 to 120,000 m/z, and the mass regions from 70,000 to 100,000 m/z that contained +1 ions of shACE2 are depicted. Values on the top of the peaks are centroids of the peaks that correspond to the molecular weight of each shACE2 protein. **d, e** Size differences in the average molecular weight between shACE2 wt and the glycan deletants for N-to-S substitution (**d**) and N-to-A substitution (**e**). Representative data and images from two experiments with different protein preparations are shown.

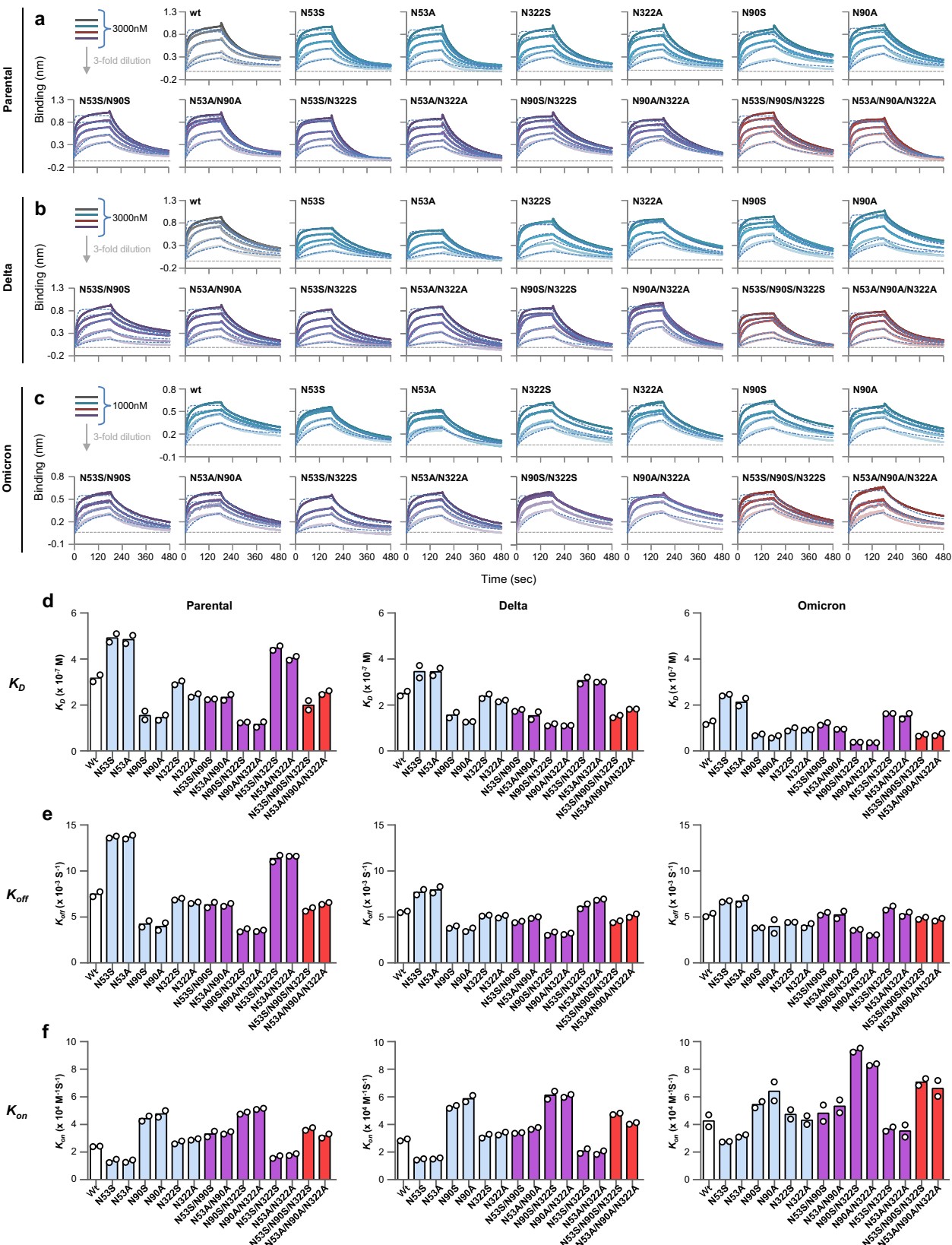

Parental virus (Fig. 4d, e). Furthermore, the N to S variant again showed similar changes in virus entry of the two lineages as did the N to A variants. These results indicate that the altered affinity resulting from hACE2 glycan variations directly contributed to CoV2 uptake during adsorption on host cells.

**Glycosylation-optimized shACE2 acts as a superior decoy receptor against Parental, Delta and Omicron.** We measured the neutralizing activities of shACE2-wt, the N90A single deletant, the N90A/N322A double deletant and the N53A/N90A/N322 triple deletant against Parental, Delta and Omicron. A virus focus

**Fig. 3 Profiles of CoV 2 Parental, Delta and Omicron S-RBD binding to hACE2 glycan deletants. a–c** Bio-layer interferometry binding analysis of the shACE2 glycan deletants to immobilized CoV2 Parental S-RBD (**a**), Delta S-RBD (**b**), and Omicron S-RBD (**b**). The experiments were repeated twice with different protein preparations and one representative set of curves is shown. Raw sensor grams and fitting curves are shown in color and with dots, respectively. Three-fold serial dilutions from 3000 or 1000 nM hACE2 added on to the immobilized CoV2 RBD. **d–f** Summary of hACE2 variant binding profiles, $K_D$ (**d**), on-rate (**e**) and off-rate (**f**) to CoV2 Parental S-RBD, Delta S-RBD, and Omicron S-RBD. Kinetic data were globally fitted with a 1:1 binding model with all binding curves that matched the theoretical fit with $R^2 > 0.95$ and an average $K_D$, off-rate and on-rate obtained from duplicates with different protein preparations are shown.

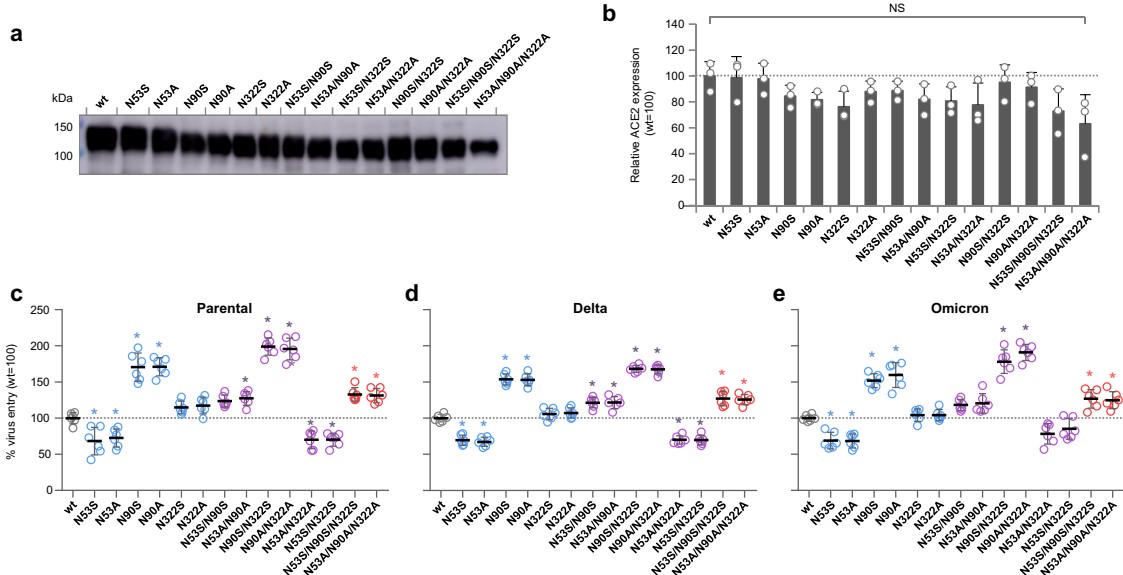

**Fig. 4 hACE2 glycan deletants affect virus entry during the adsorption of CoV2 Parental, Delta and Omicron. a**, **b** HEK293 cells with submarginal levels of endogenous hACE2 were transfected with hACE2 expression plasmids with the indicated glycan deletions, and cell surface fractions were then analyzed by western blotting using anti-hACE2 antibody. **a** Representative image of the western blots. **b** Expression of hACE2 glycan deletants relative to wt. Each data point is the mean ± SD from three independent experiments. NS indicates not significant. **c–e** HEK293 cells ectopically expressing hACE2 wt or glycan deletants were inoculated with 100 focus-forming units of CoV2 Parental (**c**), Delta (**d**), or Omicron (**e**) that were equilibrated by focus-forming assays on hACE2 wt-expressing HEK293 cells. After virus adsorption for 1 h, cells were washed and overlaid with 1% methylcellulose in culture medium. At 16 h post infection, cells were fixed and immunofluorescence assays conducted using anti-CoV2 nucleoprotein antibody. Visualized foci, each of which were derived from infection with a single virion, were counted, and the number of foci relative to wt was calculated as a surrogate measure of virus entry efficacy. Each data point is the mean ± SD from six independent experiments. Statistically significant differences compared to hACE2-wt are shown (*$P < 0.05$).

reduction neutralization test (FRNT) showed that shACE2-wt and the glycan deletants neutralized all three variants with low $FRNT_{50}$ values (0.64–6.4 µg/ml) (Fig. 5a–c). The three glycan deletants had superior neutralizing activities against the viruses relative to the wt form, with profiles indistinguishable from one another. Of the three glycan deletants, the N90A/N322A double deletant had the highest neutralizing activity with the lowest $FRNT_{50}$ values, whereas the N90A single deletant and the N53A/N90A/N322A triple deletant were less effective than the double deletant. These data indicate that modifying the ACE2 glycosylation pattern in a specific manner resulted in the development of shACE2 to an ACE2 decoy with improved neutralizing activity against multiple CoV2 lineages.

**shACE2 glycans at N53, N90 and N322 have different roles for ACE2-RBD binding based on their distinct contact dynamics for the RBD**. Our experimental results clearly demonstrated that the glycan at N53 contributed positively to the shACE2-RBD interaction, whereas the glycan at N90 negatively impacted the binding, and the effect of the glycan at N322 was moderate. To directly visualize the interactions between each glycan in shACE2 and S-RBD, we performed triplicate 100 ns MD simulations of the glycosylated shACE2-RBD complex (Supplementary Fig. 1 and Supplementary Movie 1). We first assessed the number of contact points between each glycan and the RBD (Fig. 6). As expected, the

number of contacts of the glycan at N90 was the highest whereas N53 contacts were very rare, and N322 was intermediate. These results suggest that the negative interactions involved in the glycans at N90 and N322, as observed in our BLI measurements (Fig. 3), were realized through non-covalent direct contact between them and the RBD. To explore the positive effect of the glycan at N53 on hACE2-RBD binding, we also performed MD simulations of the hACE2 without S-RBD. By comparing changes in solvent-accessible surface areas with or without the glycans, we quantified each glycan's coverage of the RBD binding site on ACE2 (Fig. 7). Coverage of the glycan at N90 was the highest, followed by N53. Considering the respective negative and positive impacts of the glycans at N90 and N53 on RBD binding (Fig. 3), our computational results imply that the glycan at N90 is highly flexible, such that it can interact with ACE2 itself when in close proximity to the RBD and/or interfere with RBD binding through steric hindrance. However, although the glycan at N53 rarely contacted the RBD (Fig. 6), the smaller coverage of the RBD binding site on ACE2 might have led to recruiting the interaction between hACE2 and S-RBD, thus improving the binding affinity.

## Discussion

The effects of glycosylation on CoV2-S and hACE2 on their binding to one another have been predicted in multiple MD

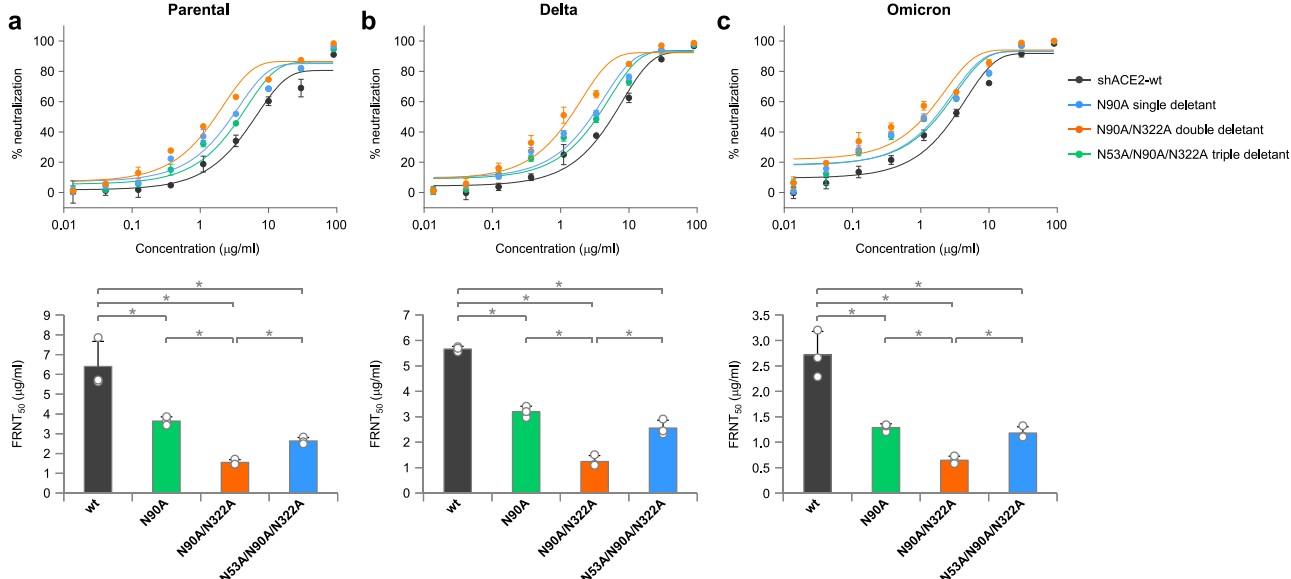

**Fig. 5 Glyco-optimized shACE2 forms possess improved neutralizing activities against CoV2 Parental, Delta and Omicron.** (**a–c**, upper) Neutralization kinetics against CoV2 Parental (**a**), Delta (**b**), and Omicron (**c**) using shACE2-wt and the indicated glyco-modified shACE2 variants at serially diluted concentrations (0.0137 to 90 μg/ml). (**a–c**, lower) Neutralizing activities of shACE2 are shown as the 50% focus reduction neutralizing titers ($FRNT_{50}$) calculated by fitting the kinetic data of the upper panel. Each data point is the mean ± SD from three independent experiments. Statistically significant differences are shown as *$P < 0.05$.

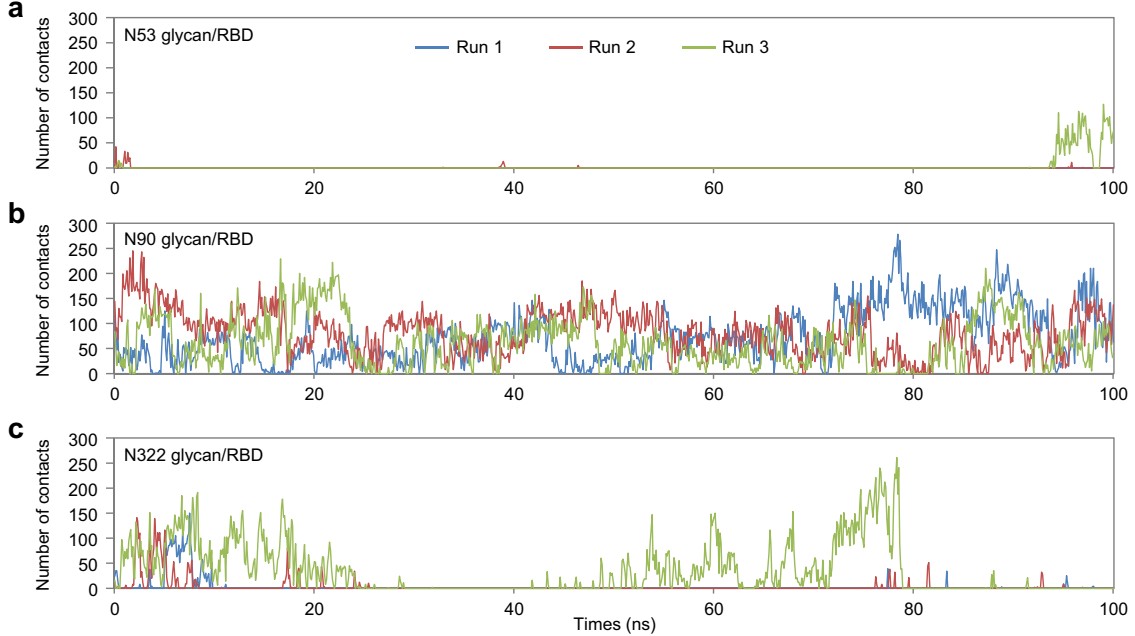

**Fig. 6 ACE2 glycan contacts in the ACE2-RBD complex.** Time trace of the number of contacts between ACE2 glycans at N53 (**a**), N90 (**b**), and N322 (**c**) and S-RBD residues in three simulation runs.

simulations[28–31]. However, few reports have measured the actual effects of hACE2 glycosylation on CoV2 S-RBD binding affinity and inhibitory activity of CoV2 infection except one previous study assessing both of these effects using the CoV2 parental strain[41]. No studies have assessed their effects on subsequent viral uptake into host cells by infection. Experimental validation is critical because glycan compositional diversity and conformational flexibility may pose a challenge to simulations. In the present study, we validated the effect of glycosylation as individual hACE2 sites on these two parameters (binding profile and virus entry) using a series of hACE2 mutants lacking *N*-glycans at

three positions (N53, N90 and N322) that are structurally close to the interface with CoV2-S. We showed that the modification of these hACE2 *N*-glycans contributed to distinct changes both in CoV2-S affinity and viral entry efficacy during adsorption across the CoV-2 Parental, Delta and Omicron strains in a site-specific manner. This approach revealed that each of the three *N*-glycans had positive or negative effects on the binding affinity of CoV2-S and virus uptake efficacy, up to about a two-fold increase or decrease. N90 glycan deletion appreciably increased CoV2-S binding and virus uptake. This indicated that N90 glycan hampers hACE2-S-RBD interactions, in accordance with the MD

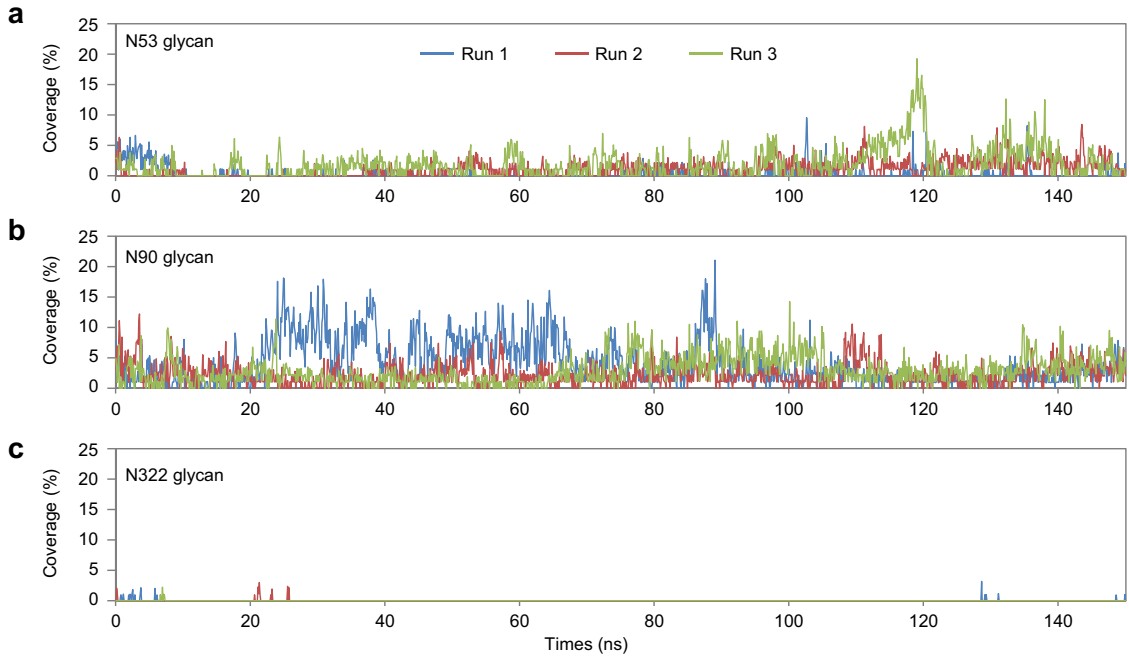

**Fig. 7 Coverage of the RBD binding site by ACE2 glycan at N53, N90 and N322.** Time trace of the coverage of ACE2 glycans at N53 (**a**), N90 (**b**), and N322 (**c**) with the putative RBD binding site on ACE2 in three simulation runs.

simulations[31,41]. Conversely, the N53 glycan deletion reduced CoV2-S binding and virus entry 2-fold, showing that N53 glycan supported the interaction between these two molecules, and facilitated viral entry. N53 glycan is expected to indirectly stabilize the hACE2-RBD complex[30], consistent with our data, although less frequent contact with CoV2-S was predicted by simulations compared to glycans on N90 and N322[29,31]. In contrast, N322 glycan had a weaker effect and had only a slightly negative role in binding and virus uptake. N322 glycan, along with N90 glycan, are expected to frequently contact CoV2-S, to enhance binding, according to several MD calculations[29,31]. Our actual data did not match these expectations. However, one of the models also indicated that N546 glycan moves the N322 glycan away from the RBD in simulation trials, prohibiting its interaction with the RBD[31]. Our results suggested that such motility actually occurs more frequently than expected. Moreover, our interpretation of the negative N322 glycan contribution to hACE2-RBD binding was in accordance with a recent MD study predicting its unfavorable impact on binding due to steric restriction and a loss of conformational freedom[41].

When the three glycans were deleted in different combinations, they cooperatively acted on CoV2-S binding and viral entry. Thus, these parameters were changed additively in some combinations and negatively in others. In fact, the binding-enhancing combinations N90A/N322A and N90S/N322S conferred the highest values of CoV2-S affinity and virus uptake in this study, whereas the lowest resulted from N53A or N53S alone, which had a reducing effect specific for these positions. Conversely, the triple N53A/N90A/N322A and N53S/N90S/N322S combinations only moderately increased binding as a result of offset effects caused by the three glycan deletions. These results indicate that for the optimization of hACE2-Spike binding affinity it is necessary to modify (add or delete) ACE2 glycosylation site-selectively, but not prevent glycosylation in its entirety. Our results suggest that multiple glycosylation sites on hACE2 involve cooperative hACE2-S RBD interactions and virus entry. In fact, a recent study reported that when glycosylations at N90 and N322 were deleted, as selected based on a computer simulation, the ACE2 variant

acted as a powerful decoy that exerted a higher competitive effect that inhibited CoV2 infection[41], in accordance with our results. However, in that report, the effect of N53 glycosylation was not evaluated because it was predicted to have less frequent contact with RBD in a computer simulation.

Comparing wt hACE2 with Parental, Delta and Omicron RBDs, their affinities were lowest for Parental, intermediate for Delta and highest for Omicron. Of the hACE2 glycan variants, the highest affinity was observed for the interaction between the Omicron RBD and hACE2-N90S(A)/N322S(A), whereas the lowest was seen between the Parental RBD and hACE2-N53S(A). Nonetheless, all three viruses exhibited almost identical variation patterns to these hACE2 glycan variants, relative to hACE2 wt. In particular, Omicron had a pattern of variation indistinguishable from the Parental virus, despite containing as many as 15 mutations in the RBD compared to Parental. This suggests the requirement for substantial conservation of the interaction between CoV2 (parental and variant lineages) and hACE2 glycans located at the interface.

Previous MD analyses simulated ACE2-Spike interactions by adding the glycan structures that were arbitrarily selected or identified as the most representative by glycomics on each glycosylation site[29–31]. These simulations have led to the extremely rapid acquisition of information contributing to a better understanding of hACE2 and Spike interactions. However, the effects of ACE2 glycans on binding and virus entry have only been analyzed physically by deleting whole glycans using enzymatic treatment or CRISPR-Cas9[32,34]. Our data delineating the site-specific contributions of hACE2 glycans to CoV-S binding affinity suggest the need to complement the preceding in silico data by actually evaluating the effect of individual glycans on the binding properties in reality.

In conclusion, our data provide important insights into designing improved therapeutic shACE2 decoys. Additionally, our findings of the high degree of conservation of the interaction between ACE2 glycans and multiple CoV2 lineages indicate the potential for improving the inhibitory activity of glycan-optimized shACE2 against new CoV2 variants that may emerge in the future.

## Methods

**Cell culture**. HEK293 cells (human embryonic kidney cell line) and Vero cells (African green monkey kidney cell line) were obtained from the RIKEN BioResource Center Cell Bank. Vero/TMPRSS2[43] were obtained from the Japanese Collection of Research Bioresources Cell Bank. 293T cells were maintained in Dulbecco's modified Eagle's Medium (DMEM) containing 10% fetal calf serum (FCS). Vero cells and Vero/TMPRSS2 cells were maintained in DMEM containing 5% FCS and 1 mg/ml G418 (Invivogen) was added to the growth medium for Vero/TMPRSS2 cells.

**Viruses**. The parental Wuhan strain (JPN/TY/WK-521/2020), the Delta (B.1.617.2 lineage) strain (Japan/TY11-927/2021) and the Omicron (B.1.1.529.2/BA.2 lineage) strain (Japan/TY40-385/2022) of CoV2 were kindly provided by the National Institute of Infectious Diseases, Tokyo, Japan. CoV2 was propagated once in Vero/TMPRSS2 cells in DMEM-F12 medium containing 0.2% bovine serum albumin (BSA) at 37 °C. Virus titration was performed by measuring focus-forming units (FFU) in focus-forming assays[44] on Vero cells.

**Plasmids**. Full length or the ectodomain of hACE2 with or without Twin-Strep-Tag at the C-terminus was amplified by PCR using specific forward primers containing an *Xho*I restriction site and reverse primers containing a *Kpn*I restriction site. The amplified fragments were digested with enzymes and cloned into polymerase II-driven pcXN2 plasmids[45].

**Amino acid substitution mutagenesis of human ACE2**. The potential glycosylation motifs at residues 53–55, 90–92 and 322–324 on hACE2 are Asn-Ile-Thr, Asn-Leu-Thr and Asn-Met-Thr, respectively. For loss of glycosylation, Asn to Ser or Ala mutation of the first amino acid of each motif was introduced into hACE2 wt using KOD-Plus mutagenesis kits (TOYOBO) according to the manufacturer's instructions and as described previously[46]. All the mutations were confirmed by sequence analysis.

**Expression and purification of recombinant human ACE2 proteins**. HEK293 cells were transfected with pcXN2 plasmids encoding the ectodomain of hACE2 wt or glycan deletants fused with Twin-Strep-Tag at the C-termini. Cell culture supernatants were harvested 24 h post transfection and centrifuged at 1500 rpm for 10 min to remove cell pellets. The precleared supernatants were then incubated with BioLock (IBA) for 15 min at room temperature (RT), followed by centrifugation at 9000 rpm for 5 min. hACE2 proteins were affinity-purified using Strep-Tactin XT Super flow high-capacity resin (IBA). Purified proteins were buffer exchanged into PBS using Slide-A-Lyzer™ G2 Dialysis Cassettes (10 K MWCO) (Thermo Fisher). Concentrations of purified proteins were measured using Pierce BCA protein assay kits (Thermo Fisher). The purity of purified proteins was assessed by SDS-PAGE and Coomassie Blue staining.

**Viral infection and immunofluorescence assay**. Twenty-four hours after transfection with pcXN2 expressing full-length hACE2 wt or glycan deletants, HEK293 cells were infected with 100 FFU CoV2 Parental, Delta or Omicron viruses. After incubation at 37 °C for 1 h, the cells were washed and overlaid with 1% methylcellulose in modified Eagle's medium containing 0.2% bovine serum albumin (BSA). At 16 h post infection, cells were washed with PBS and fixed in 4% paraformaldehyde in PBS. Immunofluorescence assays were performed with anti-CoV2 Nucleocapsid antibody (GeneTex) or anti-SARS-CoV-1/2 Nucleocapsid antibody 1C7C7 (Sigma-Aldrich) and Alexa-fluor 488 secondary antibody. Foci, each of which was derived from infection with a single virion, were counted using an inverted fluorescence microscope (ECLIPSE Ti2, Nikon).

**Biolayer interferometry analysis**. Biolayer interferometry was performed using the BLItz System (SARTORIUS/ForteBio) with HEK293-derived recombinant monomeric CoV2 S-RBD fused with human Fc (hFC). A 1.5 μg/ml concentration of S-RBD from the Parental virus with hFc (SinoBiological, 40592-V02H), Delta S-RBD fused with hFc (L452R and T478K that are characteristic of the B.1.1.617.2 lineage) (SinoBiological, 40592-V02H3) and Omicron S-RBD with hFc (G339D, S371L, S373P, S375F, K417N, N440K, G446S, S477N, T478K, E484A, Q493R, G496S, Q498R, N501Y and Y505H that are characteristic of the B.1.1.529.2/BA.2 lineage) (R&D System, 11057-CV) was immobilized on anti-human IgG-Fc (AHC)-coated biosensors for 90 s. The baseline interference phase was obtained by measurements taken for 30 s in the reaction buffer (RB: 1x PBS and 0.05% Tween-20). The sensors were subjected to the association phase for 180 s in wells containing 3-fold serially diluted wt hACE2 or hACE2 glycan deletants that were generated as above in RB. The sensors were then immersed in RB for 300 s as the dissociation step. The mean $K_{on}$, $K_{off}$ and apparent $K_D$ values of SARS-CoV-2 RBD binding affinities for hACE2 were calculated from all the binding curves based on their global fit to a 1:1 Langmuir binding model (ForteBio Data analysis 9.0 software) with an $R^2$ value of >0.95.

**Assessment of cell surface expression levels of human ACE2**. Quantification of ectopic hACE2 expression on the surface of HEK293 cells was achieved using Pierce Cell Surface Biotinylation and Isolation kits (Thermo Fisher), according to the manufacturer's instructions. Briefly, HEK293T cells cultured in 10 cm dishes were transfected with equal amounts of pcXN2 plasmid expressing full-length hACE2 wt or glycan deletants. At 24 h post transfection, cells in each dish were washed with 4 ml of ambient temperature PBS and biotinylated using 5 ml of Sulfo-NHS-SS-Biotin Solution for 10 min at RT. After removing the labeling solution, the biotin-labeled cells were washed twice with 10 ml ice-cold TBS. The cells were then scraped off, transferred into 1.5 ml micro-tubes and lysed with 180 μl of Lysis Buffer for 30 min at RT. The lysate was clarified by centrifugation at 15,000 rpm for 5 min at 4 °C to remove the cell debris (Input samples). To normalize the amount of Input samples, the protein concentration of clarified cell lysate from each dish was measured using Pierce BCA Protein Assay Kits (Thermo Fisher). An equal amount of Input samples was mixed with 250 μl slurry of NeutrAvidin resin for 30 min at RT, washed with Wash Buffer and then eluted with 200 μl of DTT solution. Extracellular levels of hACE2 wt and glycan deletants on the cells surface in Input samples were analyzed by western blotting as described below.

**Western blotting**. The cell surface fractions prepared as described above were mixed with sample buffer and applied to western blotting, as described previously[47]. Briefly, the samples were resolved by SDS-PAGE and transferred onto polyvinylidene difluoride membranes (Millipore). Western blotting was performed with anti-hACE2 antibody (GeneTex Cat #: GTX101395) and HRP-conjugated secondary antibody (Jackson ImmunoResearch Cat #: 711-035-152). The bands were visualized with the Amersham ECL Select Western blotting detection reagent and an Amersham Imager 680 (GE Healthcare). Band intensities were quantified by Amersham Imager 680 Analysis software (GE Healthcare).

**Matrix-assisted laser desorption/ionization time-of-flight mass spectrometry**. Protein samples, each containing 10–20 μg of purified hACE2 protein in PBS, were desalted with GL-TIP SDB (GL-Science) according to the manufacturer's protocol and analyzed by Matrix-assisted laser desorption/ionization time-of-flight mass spectrometry (MALDI-TOF MS) (Autoflex maX, Bruker) operated in the linear mode with a detection range from 15,000 to 120,000 m/z. Sinapinic acid was used as the matrix, and calibration was carried out with + 1 and + 2 ions of phosphorylase b from rabbit muscle (average mass: 97,429.4, Sigma-Aldrich). Peak m/z values were calculated as centroids using FlexAnalysis software (Bruker).

**Focus reduction neutralization test**. Neutralization activities against CoV2 were measured by a FRNT as described previously[48]. Briefly, serial dilutions of shACE2-wt and the indicated glycan deletants (starting concentration, 90 μg/ml) were mixed with 100 FFU of virus and incubated for 1 h at 37 °C. The mixture was then adsorbed on Vero E6 cells in 96-well plates for 1 h at 37 °C. The cells were washed and overlaid with 1% methylcellulose in modified Eagle's medium containing 0.2% bovine serum albumin. After culturing for 12 h at 37 °C, the cells were washed with PBS and fixed in 4% paraformaldehyde. Immunofluorescence assays and foci counting were performed as described above. The results are expressed as the 50% focus reduction neutralization test titer ($FRNT_{50}$), calculated using Origin 9.1 (OriginLab).

**Molecular dynamics simulations**. MD simulations of hACE2 wt in complex with CoV2-S RBD (Parental virus), as well as hACE2 alone, were performed using GROMACS 2018[49] with the CHARMM36m force field[50]. Initial structures of fully-glycosylated hACE2 and CoV2-S RBD, where the most frequent N-glycans were added to N53, N90, N103, N322, N546 of hACE2 and N331, N343 of CoV2-S RBD based on mass spectrometry data[26], were retrieved from the COVID-19 Archive of the CHARMM-GUI[51]. A zinc ion was also added to the coordinated site in the glycosylated hACE2. Using the CHARMM-GUI[52], the initial structures were solvated with TIP3P water in a rectangular box such that the minimum distance to the edge of the box was 10 Å under periodic boundary conditions. Na and Cl ions were added to neutralize the protein charge, then further ions were added to mimic a salt solution concentration of 0.15 M. Each system was energy-minimized for 5000 steps and equilibrated with NVT ensemble at 298 K for 250 ps. A further production run was performed for 100 ns with NPT ensemble. A cutoff distance of 12 Å for Coulomb and van der Waals interactions was used. Long-range electrostatics were evaluated using the Particle Mesh Ewald method[53]. The LINCS algorithm was used to constrain bonds involving hydrogen atoms[54]. The time step was set to 2 fs throughout the simulations. A simulation was repeated 3 times for each system, and snapshots were saved every 100 ps. All the trajectory analyses were performed using the GROMACS packages, and UCSF Chimera[55] was used to visualize the MD trajectories and to render the molecular graphics.

A protein residue and a glycan were considered to be in contact when at least one heavy-atom pair was within 4.0 Å. To compute the solvent accessible surface

area, we used the probe size of 5 Å, which was expected to identify regions shielded by the glycan without direct interactions between the glycan and the protein[31].

**Statistics and reproducibility.** Data analyses were performed using GraphPad Prism Version 6 software (GraphPad Software). Statistically significant differences between virus pairs were determined by ANOVA with Tukey's multiple comparison test. Data are presented as the means ± SD. Reproducibility was confirmed by performing two independent experiments with different protein preparations (for Figs. 2 and 3) or three independent replicates (for Figs. 4–7 and Supplementary Fig. 1) as described in the figure legend.

**Reporting summary.** Further information on research design is available in the Nature Research Reporting Summary linked to this article.

## Data availability

Crystal structures were obtained from Protein Data Bank with accession code PDB ID: 6VW1. All source data underlying the graphs and charts presented in the figures are presented in Supplementary Data 1. Uncropped and unedited blot images are also provided in Supplementary Fig. 2.

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

## Acknowledgements

We thank all authors who have kindly deposited and shared SARS-CoV-2 genome data on GISAID. SARS-CoV-2 Parental strain, Delta strain and Omicron strains were obtained from the National Institute of Infectious Diseases, Japan. This study was supported by JST-CREST JPMJCR15F4 (to K.M.), the JST-MIRAI Program JPMJMI19D4 and JPMJMI22D2 (to K.M.), partially by JSPS Grants-in-Aid for Scientific Research, 22H02879 (Y.W.), 20K21723 (to Y.W.), 20KK0224 (to Y.W.) and 21K15455 (to Y.A.), and partly by grants from the Takeda Science Foundation (to Y.W.), the Uehara Memorial Foundation (to Y.A.), SEN-SHIN Medical Research Foundation (to Y.W.), and the Chemo-Sero-Therapeutic Research Institute (to Y.W.). The supercomputing resources in this study were provided in part by the Human Genome Center at the Institute of Medical Science, The University of Tokyo, and by Research Center for Computational Science, Okazaki, Japan (Project: 22-IMS-C091).

## Author contributions

A.I., Y.A., and Y.W. performed the experiments. D.K. performed in silico experiments. N.O. performed MALDI-TOF MS analyses. T.O., S.U., S.N., T.D., Y.S., T.N., K.M., and Y.W. interpreted the results. Y.A., and Y.W. conceptualized the study and designed the experiments. A.I., Y.A., D.K., and Y.W. wrote the manuscript. All authors reviewed and proofread the manuscript.

## Competing interests

The author S.U. is an employee of Murata Manufacturing Co., Ltd. The other authors declare no competing interests exist.
