## [Peer Review File · Communications Biology]

Reviewers' comments:

Reviewer #1 (Remarks to the Author):

ACE2 (angiotensin converting enzyme 2) is the cellular receptor of SARS-CoV-2 and soluble ACE2 is evaluated as a treatment option.

In this study, Isobe and colleagues functionally analyzed the role of three ACE2 glycosylation sites that are located close to the ACE2/SARS-CoV-2 spike binding interface, and further investigated whether soluble ACE2 versions with deleted glycosylation sites may display increased antiviral activity. By mutating the asparagine residues within glycosylation motifs at positions 53, 90 and 322 to either alanine or serine, the authors show that glycosylation at position N53 has a positive effect on ACE2/SARS-CoV-2 spike interaction, while glycosylation at position N322 and especially N90 negatively impact ACE2/SARS-CoV-2 spike interaction. The binding data were further shown to correlate with infectivity data with different authentic SARS-CoV-2 isolates. Moreover, the positive or negative impact of the respective mutations was similar for parental SARS-CoV-2, Delta variant and Omicron variant, suggesting that the impact of the three ACE2 glycosylation sites is conserved among multiple (if not all) SARS-CoV-2 lineages. Finally, the authors designed shACE2 versions in which the glycosylation site at position N90 was deleted either alone or jointly with the glycosylation sites at position N322 or N53/N322, and showed that these shACE2 versions displayed higher anti-SARS-CoV-2 activity than wildtype hACE2.

In sum, this is an interesting straightforward study that extends our understanding on the importance of ACE2 glycosylation on ACE2/SARS-CoV-2 spike interaction, and provides guidance for further optimization of soluble ACE2 for therapeutic use. The experimental set-up selected is appropriate to tackle the research hypothesis and the data support the authors' conclusions. I have only a few minor points that should be easy to address.

Minor points:

- Line 66-67: Mutation D614G does not increase ACE2 affinity but augments infectivity by increasing the stability of viral particles (PMID: 33106671) and inducing structural changes that favor the receptor-binding domain to adopt the so-called "up-conformation" required for ACE2 binding (PMID: 33417835).
- Line 67: Please name the exact mutation for position 439 (maybe N439K?).
- Line 67: Spike mutations N501Y, S477N and N439K are not only escape mutations but also increase ACE2 affinity (PMID: 34818667, PMID: 34435953, PMID: 33621484)
- Figure 5: There seems to be a labeling error in the lower panels. I think the label "N53A/N90A" should read "N90A/N322A"?
- Introduction/Discussion: multiple efforts have been spent by several groups to optimize shACE2 for increased inhibitory activity, including the addition of multimerization domains (e.g., PMID: 34012108) and introduction of mutations found in ACE2 orthologues of different animal species (e.g., PMID: 33836016). The author should also mention these strategies of shACE2 optimization in the introduction and/or discussion.

Reviewer #2 (Remarks to the Author):

In the present manuscript, Isobe A. et al., report on the site-specific impacts of hACE2 N-glycosylation (at positions N53, N90 and N322) on the interaction with three variants of the SARS-CoV-2 spike protein (parental Wuhan strain, variants Delta and Omicron). In line with previous publications, the authors find that deletions of protein N-glycosylation at N90 (and to a lesser extent also at N322) results in an increase of the binding affinity to RBD and confirm this observation for all RBD variants tested in this study. Deletions at N52 have a detrimental effect on hACE2 binding to all RBD variants tested. Combinations of deletions were found to have an additive effect. Furthermore, again in line

with previous studies, the authors show that hACE2-RBD binding affinities are highly indicative for both, viral infection rates (as measured by in vitro infection assays) and the neutralization activity of soluble hACE2 decoy receptor variants. In large parts (except for N-glycosylation deletions at N52) the present study confirms previous work, now also for two recently emerged SARS-CoV-2 spike-protein variants of concern (Delta and Omicron). The manuscript is very well written, previous work is -for the most part (see below)- appropriately cited, experiments are conceptually and scientifically sound, and the authors conclusions are well supported by the data presented. I cannot comment on the quality or merit of the molecular dynamics simulations presented in the manuscript.

Major point:

Line 225 – 227: “However, fewer reports have measured the actual effects of hACE2 glycosylation on CoV2 S-RBD binding affinity (38) and none assessed their effects on subsequent viral uptake on infection.” - This statement is not correct. For example, in reference 38 (Capraz et al., 2021; Fig. 7. Critical role of ACE2 glycosylation for SARS-CoV-2 infectivity) the effect of glyco-deletion-associated differences in hACE2-RBD binding affinities on SARS-CoV-2 infection was clearly assessed. Please change.

Reviewer #3 (Remarks to the Author):

In this study, Isobe et al. provided experimental data to evaluate the effect of hACE2 glycosylation on the interaction with SARS-CoV-2 spike, focusing on N53, N90 and N322. They firstly examined ACE2/spike binding using a BLI assay, showing that removal of N53-linked glycan reduced affinity, whilst removal of N90- and/or N322-linked glycans each enhanced affinity, which is, surprisingly, opposed to various previous studies where the N322-linked glycan was predicted to aid RBD binding (thus removal would reduce affinity); a triple deletion mutant showed a reduction in affinity. They further validated their findings by showing that enhanced affinity of the mutants was translated into enhanced viral entry and improved neutralizing activity as a decoy.

The results sound robust but there are a few comments:

(1) Line 32-35: “Our study provides insight helpful for designing optimized anti-spike agents and also presents evidence for predicting ACE2 binding properties of any new SARS-CoV-2 variants that may emerge during virus evolution in future.”

Line 87 -89: “Therefore greater understanding of the interactions between CoV2-S and hACE2 is required to predict the binding phenotype of CoV2 variants that will appear in the future.”

Line 294-297: “Additionally, our findings of the high degree of conservation of the interaction between ACE2 glycans and multiple CoV2 lineages provide suggestive evidence for predicting hACE2 binding properties by new CoV2 variants that may emerge in future.”

Comment: These over-statements on binding and evolution predictions are irrelevant to this study.

(2) Line 67-68: “In contrast, N501Y, S477N and 439 on the RBD affects recognition by serum antibodies.”

Comment: The effects need to be clarified.

(3) Line 83-87: “CoV2 vaccines based on novel messenger RNA technology have been licensed for human use. Additionally, hACE2 peptide-mimics which block CoV2 infection have been investigated as therapeutic agents. Use of these interventions results in a higher selective pressure on the virus and may accelerate its evolution, especially for Spike, as has been observed for antibodies.”

Comment: I don't think these are true regarding the higher selective pressures. Also, if the viruses

escapes the ACE2 decoy then it will be losing affinity for the receptor and hence have reduced transmissibility instead.

4) Line 95-97: "Our experimental findings guide rational design of therapeutic hACE2 peptide-mimics with improved blocking efficacy by optimizing their glycosylation pattern to achieve the strongest interaction between the two molecules."

Comment: It is misleading to use 'peptide' in this study, as the soluble form of ACE2 receptor was used.

(5) Line 135-137: "Omicron are shown in Figs. 3a, 3b and 3c respectively. KD, on-rate and off-rate were calculated by global fitting to a 1:1 steady-state binding model and expressed as the average of two analytical repeats (Figs. 3d-f)."

Comment: No fitting of the binding curves was shown on the figures.

(6) Line 138-139: "The shACE2 glycan variants appreciably changed the binding affinity of Parental RBD up to about 2-fold higher or lower than wt shACE2 (Fig. 3d left)."

Comment: Appreciably / appreciable should not be used unless statistical data is provided.

(7) Line 143-144: "Combining each glycan deletion resulted in an effect on RBD binding in an additive or offsetting manner:"

Comment: Combined mutations didn't lead to further reduced affinities compared to the single mutations.

(8) Line 170-172: "The N90A deletant increased the entry of Parental virus during adsorption to 1.71-fold the wt, whereas the N53A deletant reduced it to 0.72-fold, the lowest in this study (Fig. 4c)"

Comment: It should be reduced/increased by X-fold.

(9) Line 29-30: "These glycosylation deletions caused distinct site-specific changes of interactions with spike and acted in a delicately balanced manner".

Line 265-267: "Our results suggest that multiple glycosylation sites on hACE2 contribute to hACE2-S RBD interactions and virus entry in a balanced manner."

Comment: I don't think the data supports the statement for a 'balanced manner'. For this, the authors need to show evidence that the glycans have additional effects other than on affinity.

(10) Line 293-294: "In conclusion, our data provide important insights into designing improved therapeutic ACE2 peptide decoys and also SARS-CoV-2 vaccines.

Comment: Again, 'peptide' should not be used here. Also, I can't see it has anything to do with vaccine design.

Authors' reply to the reviewers' comments

We wish to express our sincere appreciation to the reviewers for their insightful comments on our manuscript. Their comments have helped us considerably in revising and planning to improve our paper.

In our response to each of the reviewers' comments below, the comment is in italics and is followed by our response in regular font.

Reviewer #1

We appreciate that the reviewer noted: "this is an interesting straightforward study that extends our understanding on the importance of ACE2 glycosylation on ACE2/SARS-CoV-2 spike interaction, and provides guidance for further optimization of soluble ACE2 for therapeutic use." and "the experimental set-up selected is appropriate to tackle the research hypothesis and the data support the authors' conclusions." The reviewer also expressed concerns about the issues below. In response, we have made revisions in accordance with the reviewer's comments as follows:

Minor comments:

1) Lines 66-67: Mutation D614G does not increase ACE2 affinity but augments infectivity by increasing the stability of viral particles (PMID: 33106671) and inducing structural changes that favor the receptor-binding domain to adopt the so-called "up-conformation" required for ACE2 binding (PMID: 33417835).

As suggested by the reviewer, we have corrected these lines and also cited the references that the reviewer suggested in the revised manuscript (page 3, lines 60-61) to more carefully explain the scientific background related to our study.

2) Line 67: Please name the exact mutation for position 439 (maybe N439K?)

As suggested by the reviewer, we have corrected "439" to "N439K" in our revised manuscript (page 3, line 62). Thank you for pointing this error out.

3) Line 67: Spike mutations N501Y, S477N and N439K are not only escape mutations but also increase ACE2 affinity (PMID: 34818667, PMID: 34435953, PMID: 33621484).

As suggested by the reviewer, we have corrected these lines and also cited the references suggested by the reviewer in our revised manuscript (page 3, lines 61-63).

4) Figure 5: There seems to be a labeling error in the lower panels. I think the label “N53A/N90A” should read “N90A/N322A”?

As suggested by the reviewer, we have corrected the labeling of “N53A/N90A” to N90A/N322A in Figure 5 lower panels of our revised manuscript. Thank you for pointing this out.

5) Introduction/Discussion: multiple efforts have been spent by several groups to optimize shACE2 for increased inhibitory activity, including the addition of multimerization domains (e.g., PMID: 34012108) and introduction of mutations found in ACE2 orthologues of different animal species (e.g., PMID: 33836016). The author should also mention these strategies of shACE2 optimization in the introduction and/or discussion.

We totally agree with the reviewer’s comment. As suggested, we have added the relevant information and reference papers suggested to the introduction of our revised manuscript (page 4, lines 80-84) to more precisely explain the scientific background related to this study.

Reviewer #2

We appreciate that the reviewer noted: “the manuscript is very well written, previous work is appropriately cited, experiments are conceptually and scientifically sound, and the authors conclusions are well supported by the data presented.” The reviewer also expressed concerns about the issues below. In response, we have made revisions in accordance with the reviewer’s comments as follows:

Major comment:

1) Lines 225–227: “However, fewer reports have measured the actual effects of hACE2 glycosylation on CoV2 S-RBD binding affinity (38) and none assessed their effects on subsequent viral uptake on infection.” This statement is not correct. For example, in reference 38 (Capraz et al., eLife 2021; Fig. 7. Critical role of ACE2 glycosylation for

SARS-CoV-2 infectivity) the effect of glyco-deletion-associated differences in hACE2-RBD binding affinities on SARS-CoV-2 infection was clearly assessed. Please change.

We totally agree with the reviewer comments. As suggested, we have corrected these lines in our revised manuscript (page 11, lines 228-232) to more carefully refer to the paper by Capraz et al., (PMID: 34927585) .

Reviewer #3

We appreciate that the reviewer noted: “the results sound robust.” The reviewer also expressed concerns about the issues below. In response, we have made revisions in accordance with the reviewer’s comments as follows:

Minor comments:

1) The following overstatements on binding and evolution predictions are irrelevant to this study.

1-1) Lines 32-35: “Our study provides insight helpful for designing optimized anti-spike agents and also presents evidence for predicting ACE2 binding properties of any new SARS-CoV-2 variants that may emerge during virus evolution in future. ”

We agree with the reviewer’s comment. We have corrected these lines in our revised manuscript (page 2, lines 33-37) to more carefully state our findings.

1-2) Line 87 -89: “Therefore greater understanding of the interactions between CoV2-S and hACE2 is required to predict the binding phenotype of CoV2 variants that will appear in the future. ”

We totally agree with the reviewer’s comment. To address this comment, we have deleted the sentence and replaced it with a new introductory description in our revised manuscript (pages 4, lines 80-91) to more carefully explain the scientific background related to this study.

1-3) Lines 294-297: “Additionally, our findings of the high degree of conservation of the interaction between ACE2 glycans and multiple CoV2 lineages provide suggestive evidence for predicting hACE2 binding properties by new CoV2 variants that may emerge in future. “

To address the reviewer's comment, we have corrected these lines in our revised manuscript (page 14, lines 299-300) to more carefully state our conclusion.

2) Lines 67-68: "In contrast, N501Y, S477N and 439 on the RBD affects recognition by serum antibodies." The effects need to be clarified.

As suggested by the reviewer, we have corrected these lines in our revised manuscript (page 3, lines 61-63) for clarify this point (please see also our response to Reviewer #1 comments 2 and 3)

3) Lines 83-87: "CoV2 vaccines based on novel messenger RNA technology have been licensed for human use. Additionally, hACE2 peptide-mimics which block CoV2 infection have been investigated as therapeutic agents. Use of these interventions results in a higher selective pressure on the virus and may accelerate its evolution, especially for Spike, as has been observed for antibodies." I don't think these are true regarding the higher selective pressures. Also, if the viruses escapes the ACE2 decoy then it will be losing affinity for the receptor and hence have reduced transmissibility instead.

We totally agree with the reviewer's comment. To address this comment, we have deleted large parts of this text and replaced it with new descriptions in our revised manuscript (pages 4-5, lines 80-91) to more precisely explain the scientific background related to this study (please also see our response to reviewer #1 comment 5).

4) Lines 95-97: "Our experimental findings guide rational design of therapeutic hACE2 peptide-mimics with improved blocking efficacy by optimizing their glycosylation pattern to achieve the strongest interaction between the two molecules." It is misleading to use 'peptide' in this study, as the soluble form of ACE2 receptor was used.

As suggested by the reviewer, we have corrected "hACE2 peptide" to "soluble form of hACE2" in our revised manuscript (page 5, line 97) to more precisely state our findings.

5) Lines 135-137: "Omicron are shown in Figs. 3a, 3b and 3c respectively. KD, on-rate and off-rate were calculated by global fitting to a 1:1 steady-state binding model and expressed as the average of two analytical repeats (Figs. 3d-f)." No fitting of the binding

curves was shown on the figures.

As suggested by the reviewer, we have included overlaid fitting curves in Fig. 3a-c of our revised manuscript (page 29, lines 671-672). In addition, we have included information that all fitting curves for calculation in this study theoretically matched $R^2 > 0.95$ in our revised manuscript (page 18, lines 373-375; page 29, lines 666-667 and lines 670-671) to show the validity of our fitting curves.

6) Lines 138-139: "The shACE2 glycan variants appreciably changed the binding affinity of Parental RBD up to about 2-fold higher or lower than wt shACE2 (Fig. 3d left)." Appreciably/appreciable should not be used unless statistical data is provided.

As suggested by the reviewer we have deleted "appreciably" and replaced it with "perceptibly" in this part of our revised manuscript (page 7, line 140) to more carefully state our results.

7) Lines 143-144: "Combining each glycan deletion resulted in an effect on RBD binding in an additive or offsetting manner." Combined mutations didn't lead to further reduced affinities compared to the single mutations.

We agree with the reviewer's comment. In this study, as pointed out by the reviewer, no synergistic effect to further reduce affinity was seen for combined glycan deletants compared to the single deletants, because only N53A/S reduced the hACE2 affinity whereas the other two deletants (N90S/A and N322S/A) increased the hACE2 affinity. To address the reviewer's comment, we included this information in our revised manuscript (page 7, lines 150-152) to more precisely state our results.

8) Lines 170-172: "The N90A deletant increased the entry of Parental virus during adsorption to 1.71-fold the wt, whereas the N53A deletant reduced it to 0.72-fold, the lowest in this study (Fig. 4c)." It should be reduced/increased by X-fold.

As suggested by the reviewer, we have corrected these lines in our revised manuscript (page 9, lines 174-177).

9) Lines 29-30 and Lines 265-267: Statements for "in a balanced manner." I don't think the data supports the statement for a 'balanced manner'. For this, the authors need to

show evidence that the glycans have additional effects other than on affinity.

We agree with the reviewer's comment. To address this comment, we have deleted the statement "balanced manner" and replaced it with more appropriate descriptions in the revised manuscript (page 2, lines 30-31; page 13, lines 269-270) to more carefully state our findings.

10) Lines 293-294: "In conclusion, our data provide important insights into designing improved therapeutic ACE2 peptide decoys and also SARS-CoV-2 vaccines." Again, 'peptide' should not be used here. Also, I can't see it has anything to do with vaccine design.

As suggested by the reviewer, we have corrected "peptide" to "soluble form of hACE2" and also deleted the statement discussing vaccine design in our revised manuscript (page 14, line 297) to more precisely state our conclusion.

REVIEWERS' COMMENTS:

Reviewer #1 (Remarks to the Author):

The authors have addressed all my points and modified their manuscript accordingly. I have no more points and recommend the manuscript for publication.

Reviewer #2 (Remarks to the Author):

The authors have carefully revised the manuscript, following the recommendations and comments of all reviewers. I endorse publication in its present form.

Reviewer #3 (Remarks to the Author):

My comments have been appropriately addressed addressed in the revised manuscript and I support the publication of this work.